# Effect of Silicon Carbide Coating on Osteoblast Mineralization of Anodized Titanium Surfaces

**DOI:** 10.3390/jfb13040247

**Published:** 2022-11-16

**Authors:** Patricia dos Santos Calderon, Fernanda Regina Godoy Rocha, Xinyi Xia, Samira Esteves Afonso Camargo, Ana Luisa de Barros Pascoal, Chan-Wen Chiu, Fan Ren, Steve Ghivizzani, Josephine F. Esquivel-Upshaw

**Affiliations:** 1Department of Dentistry, Federal University of Rio Grande do Norte, Natal 59056, RN, Brazil; 2Department of Oral Biology, University of Florida College of Dentistry, Gainesville, FL 32610, USA; 3Department of Chemical Engineering, University of Florida, Gainesville, FL 32611, USA; 4Department of Comprehensive Oral Healthy, Adams Dental School, University of North Carolina, Chapel Hill, NC 27599, USA; 5Department of Orthopaedic Surgery and Sports Medicine, College of Medicine, University of Florida, Gainesville, FL 32607, USA; 6Department of Restorative Dental Sciences, Division of Prosthodontics, College of Dentistry, University of Florida, Gainesville, FL 32610, USA

**Keywords:** nanotechnology, dental implants, biocompatible coated materials, bone mineralization

## Abstract

The objective of this study was to evaluate the influence of the titanium nanotube diameter and the effect of silicon carbide (SiC) coating on the proliferation and mineralization of pre-osteoblasts on titanium nanostructured surfaces. Anodized titanium sheets with nanotube diameters of 50 and 100 nm were used. The following four groups were tested in the study: (1) non-coated 50 nm nanotubes; (2) SiC-coated 50 nm titanium nanotubes; (3) non-coated 100 nm nanotubes and (4) SiC-coated 100 nm nanotubes. The biocompatibility and cytotoxicity of pre-osteoblasts were evaluated using a CellTiter-BlueCell Viability assay after 1, 2, and 3 days. After 3 days, cells attached to the surface were observed by SEM. Pre-osteoblast mineralization was determined using Alizarin-Red staining solution after 21 days of cultivation. Data were analyzed by a Kruskal–Wallis test at a *p*-value of 0.05. The results evidenced biocompatibility and non-cytotoxicity of both 50 and 100 nm diameter coated and non-coated surfaces after 1, 2 and 3 days. The statistical analysis indicates a statistically significant higher cell growth at 3 days (*p* < 0.05). SEM images after 3 days demonstrated flattened-shaped cells without any noticeable difference in the phenotypes between different diameters or surface treatments. After 21 days of induced osteogenic differentiation, the statistical analysis indicates significantly higher osteoblast calcification on coated groups of both diameters when compared with non-coated groups (*p* < 0.05). Based on these results, we can conclude that the titanium nanotube diameter did not play any role on cell viability or mineralization of pre-osteoblasts on SiC-coated or non-coated titanium nanotube sheets. The SiC coating demonstrated biocompatibility and non-cytotoxicity and contributed to an increase in osteoblast mineralization on titanium nanostructured surfaces when compared to non-coated groups.

## 1. Introduction

Chemical composition and topography of implant surfaces play an important role in the rate and extent of osseointegration [1]. Many mechanical and chemical surface modifications have been developed and applied to control and improve the osseointegration of titanium-based implants. Some of these include applying different surface coatings through plasma spraying, acid-etching or anodization of implant surfaces [2,3].

The study of titanium surface technologies has progressed from bioinert surfaces such as porous titanium, to bioactive surfaces including plasma sprayed hydroxyapatite, to the most recent nanostructured surfaces [4]. The bioactive nanostructured titanium surface can be prepared in the form of tubes with diameters in nanometers and lengths ranging from several nanometers to micrometers [5]. The nanostructured surfaces mimic the morphology of the external cellular membranes of the osteoblasts that surround the implant and an increased surface area provided by the nanotubes boosts the interaction between the titanium surface and the adjacent cells [4].

Nanoscale surface modifications of dental implants have been beneficial in improving the degree of osseointegration by increasing the surface roughness and changing the surface chemistry [6]. Studies have demonstrated that human cells are suitable for interacting with nanostructured surfaces and coating these surfaces with biocompatible thin layers can improve this interaction [7,8]. In addition to surface topography, different elements have been introduced to titanium surfaces for improving their osteogenic activity [9]. Studies have also assessed the influence of nanotube diameter on osteogenic activity [10,11], but consensus is lacking.

As dental implants are becoming the restorative option of choice for restoring edentulism in adults, the dental community is trying to find ways to minimize the occurrence of peri-implantitis. Peri-implantitis is defined as mucosal inflammation and deep pocketing surrounding the implant along with 2 mm or more of alveolar bone resorption after implant loading, whereas peri-mucositis consists of mucosal inflammation without the bone resorption. Peri-implantitis prevalence can be anywhere from 1.1% to 85% [12], owing partially to a lack of standardized criteria for diagnosis. A nationwide Swedish study [13] determined the prevalence of the disease to be 45% after 10 years. Although there are many unknowns with regards to disease initiation and treatment for peri-implantitis, bacteria are the major causative factor in proliferation of the disease. An ideal implant is one that promotes osseointegration and also prevents bacterial infiltration. Bacterial proliferation around the implant initiates an inflammatory process that leads to titanium corrosion [14,15], tissue inflammation, bone loss and eventually loss of the implant. SiC coatings have demonstrated anti-bacterial as well as anti-corrosive properties and are ideal to augment promotion of osseointegration by nanotubes [7,16,17,18,19,20,21,22].

Silicon (Si) has been found to localize at the active sites of calcification in young bone and is therefore suggested to be closely related to calcium at the early calcification stage [23]. Si may also affect the adhesion, proliferation and differentiation of osteoblastic cells [24]. A previous study demonstrated the effectiveness of SiC coatings on titanium nanotubes in promoting osteoblast proliferation [8]. Another study illustrated enhanced calcium deposition by osteoblasts cultured on titanium nanostructured surfaces compared with non-nanostructured surfaces [25]. However, there are no studies that show the effect of the SiC coating of titanium nanostructured surfaces on osteoblast mineralization.

The ideal implant surface should be biocompatible, non-cytotoxic, anti-bacterial, anti-corrosive and promote osseointegration by enhancing cell mineralization. In this study we evaluated the behavior of pre-osteoblasts on titanium nanostructured surfaces to analyze the influence of titanium nanotube diameters and the potential of SiC coating on cell proliferation and mineralization.

## 2. Materials and Methods

### 2.1. Experimental Design

Sixty-eight pre-fabricated anodized titanium dioxide (ATO) nanotubes on titanium foils (0.03 mm × 10 mm × 10 mm, InRedox, Longmont, CO, USA) were used in this study. These ATO nanotube foils were made through electrochemical anodization resulting in nanotube diameters of 50 ± 10 nm and 100 ± 20 nm. Nanotubes of different diameters were used to create groups of small and large nanotubes.

Four groups (n = 17) were included in this study as follows: (1) non-coated 50 nm titanium nanotubes; (2) SiC-coated 50 nm titanium nanotubes; (3) non-coated 100 nm titanium nanotubes; and (4) SiC-coated 100 nm titanium nanotubes. 

### 2.2. Coating Process

A total of 34 samples (17 titanium nanotube sheets of 50 nm plus 17 titanium nanotube sheets of 100 nm) were coated through plasma-enhanced chemical vapor deposition (PECVD, PlasmaTherm 790, Saint Petersburg, FL, USA). Prior to deposition the samples were cleaned with acetone, then rinsed with isopropyl alcohol, dried with compressed nitrogen and treated with ozone to remove surface carbon contamination.

Silicon dioxide (SiO_2_) and silicon carbide (SiC) dielectric films were applied to nanotube titanium sheets. The deposition conditions were well-calibrated and SiO_2_ was deposited followed by SiC. The temperature was maintained at 300 °C with the deposition rate at 330 Å/min for SiO_2_ and 170 Å/min for SiC. The precursors for the SiO_2_ film were 5% silane balanced (SiH_4_) in helium and nitrous oxide (N_2_O), and methane (CH_4_) and silane (SiH_4_) were the precursors for the SiC film. Samples were subjected to a thermal annealing after the coating deposition. A total coating thickness of 10 nm was observed, with 5 nm on each side of the nanotubes.

### 2.3. Surface Characterization

Coated and non-coated ATO nanotube samples were characterized by SEM and EDX in a previous paper [26] to further determine the behavior of these coatings.

By using the semi-automatic drop shape analyzer DSA100S (KRÜSS Scientific), the wettability of coated and uncoated samples surfaces was measured through contact angle measurements. After 10 μL of DI water was dropped on the nanotube sample surface, the camera began recording the drop shape. Contact angle was then calculated by the computer connected to the analyzer. A mean value was calculated for each sample after this process was repeated four times.

### 2.4. Cell Viability

Murine pre-osteoblastic MC3T3-E1 cells (ATCC, USA) were cultured in a humidified atmosphere containing 5% CO_2_ at 37 °C. Cells were maintained in α-minimum essential media (MEM-α, Gibco), supplemented with 10% fetal bovine serum and 1% penicillin and streptomycin. All experiments were performed with cells at passages lower than 10. Four titanium sheets from each group were placed on a sterile 24-well plate and sterilized using ethanol 70% for 30 min. Cells were seeded at 2 × 10^4^ cells/mL in each well and cultured on three titanium sheets of each experimental group, and onto one empty well, used as a positive control for 1, 2 and 3 days. One extra sample of each group was immersed in media (negative control) and kept under the same conditions.

Cell viability was determined using the CellTiter-BlueCell Viability Assay (Promega, Madison, WI, USA), which was used according to the manufacturer’s instructions. After 1, 2 and 3 days, the cells were detached using trypsin-EDTA for 5 min at 37 °C, then trypsin was inactivated by adding fresh culture media. Following resuspension, 100 µL from each well were aliquoted to a 96-well plate in triplicates, 20 µL of CellTiter-Blue dye was added to each well, and the plates were incubated for 4 h at 37 °C and 5% CO_2_. Fluorescence was analyzed using a spectrophotometer (SmartSpec Plus, Bio-Rad, Hercules, CA, USA) at a wavelength of 600 nm.

### 2.5. Scanning Electron Microscopy

Titanium sheets were observed under scanning electron microscopy (FEI NOVA NanoSEM 430, FEI Company, Hillsboro, OR, USA) to identify cell attachments on their surface. For SEM analysis, one sample from each group was placed on a sterile 24-well plate and sterilized using ethanol 70% for 30 min. A quantity of 2 × 10^4^ cells/well were cultivated on each titanium sheet at 37 °C in a 5% CO_2_ for 3 days.

The pre-osteoblasts that adhered to the samples were fixed in a solution of 3% glutaraldehyde (50 wt.% in H2O, CAS#111-30-8, Sigma-Aldrich, San Luis, MO, USA), 0.1 mol/L sodium cacodylate (CAS#6131-99-3, Sigma-Aldrich) and 0.1 mol/L sucrose (CAS#57-50-1, Sigma-Aldrich, San Luis, MO, USA) for 45 min. Samples were immersed for 10 min in a buffer solution of 0.1 mol/L sodium cacodylate (CAS#6131-99-3, Sigma-Aldrich, San Luis, MO, USA) and 0.1 mol/L sucrose (CAS#57-50-1, Sigma-Aldrich, San Luis, MO, USA). Samples were then processed in serial ethanol dehydrations for 10 min each (30, 50, 70 and 100%) and dehydrated in hexamethyldisilazane (HDMS, CAS# 999-97-3, Sigma-Aldrich, San Luis, MO, USA).

Titanium sheets were sputter-coated with a palladium–gold alloy (Polaron SC 7620 Sputter Coater, Quorum Technologies, Laughton, East Sussex, UK) with a thickness of 10 nm. The SEM was operated at 10 kV, spot 3.5 and images were made in 100×, 2000× and 10,000×.

### 2.6. Mineralization

Four titanium sheets from each group were placed on a sterile 24-well plate and sterilized using ethanol 70% for 30 min. A total of 2 × 10^4^ cells/well were cultured on three titanium sheets of each group and in one empty well (positive control) at 37 °C in a 5% CO_2_ for 24 h using the regular growth media. Then, the media were changed for differentiation media that were the regular growth media supplemented with 50 µg/mL of ascorbic acid and 5 mM of β-glycerophosphate. Cells were cultivated on the titanium sheets at 37 °C in a 5% CO_2_ for 21 days and the differentiation medium was replaced every 3 days. The extra sample of each group was immersed in differentiation media (negative control) and maintained at the same conditions.

Mineralization was determined using Alizarin-Red staining solution (Sigma-Aldrich, San Luis, MO, USA), following the manufacturer’s instructions. After 21 days, the medium was removed, and the cells were washed 2 times with PBS and fixed in 70% ethyl alcohol for 1 h at 4 °C. The cells were then washed 2 times with distilled water and stained in 2% Alizarin-Red solution (pH 4.2) for 15 min at 37 °C. Unbound dye was removed by washing 3 times with distilled water.

To quantify matrix mineralization, 10% cetylpyridinium chloride was added to each well and incubated for 1h to dissolve and release the calcium-bound alizarin red. After resuspension, 100 µL from each well was aliquot to a 96-well plate, in triplicates, and the absorbance of released Alizarin-Red was measured using a spectrophotometer (SmartSpec Plus, Bio-Rad, Hercules, CA, USA) at a wavelength of 600 nm.

### 2.7. Data Analysis

The quantitative data were shown as the means ± standard deviations. Statistical analysis was performed using SPSS Statistics for Windows, version 22 (SPSS Inc., Chicago, IL, USA). Data normality was assessed by skewness and kurtosis and indicated a non-normal distribution. Statistical differences were calculated using the Kruskal–Wallis test to compare the cell proliferation between the groups at 1, 2 and 3 days, the cell proliferation between 1, 2 and 3 days for each group, and the mineralization between the groups at 21 days. A *p*-value of 0.05 was considered statistically significant.

## 3. Results

### 3.1. Wettability

The wettability of the coated and non-coated samples was measured using the contact angle. Compared to the coated groups, the contact angle of the non-coated groups decreased significantly (Figure 1). Table 1 shows the measured average ± SD contact angles of sample surfaces. The contact angle measurements showed that the coated samples presented higher hydrophilicity indicating that the coating treatment improved the hydrophilicity of the samples.

### 3.2. Surface Characterization

Characterization of different sized nanotubes was performed previously and reported in a different study [26]. For the benefit of the readers to see these results, we have included them here. SEM images of coated and non-coated nanotube samples were performed (Figure 2). The nanotubes and SiO_2_/SiC-coated ATO nanotubes were examined using energy-dispersive X-ray spectroscopy (EDS) analysis to determine the composition of the surface. The results were consistent between all the samples with different diameters and thicknesses of ATO nanotubes. The representative EDX spectra are shown in Figure 2. Figure 2d shows the main elements as Ti, O, F and Al from the non-coated ATO nanotubes. Figure 2e shows additional Si elements on ATO nanotubes after the SiO_2_/SiC coating was applied.

In addition, transmission electron microscopy (TEM) was used to determine the coating morphology on the internal surface of the nanotubes. This study demonstrated that the SiO_2_/SiC coating fully covered the internal surface of the nanotubes. EDS of the cross-sectional ATO nanotubes revealed the presence of Ti, O, Si and C, as further proof of the coatings reaching the internal surface (Figure 2c).

### 3.3. Cell Viability

Our results demonstrated biocompatibility of both 50 and 100 nm diameter coated and non-coated surfaces when evaluated by the absorbance of the CellTiter-Blue assay after 1, 2 and 3 days of MC3T3 pre-osteoblasts growth on the nanotube samples. The Kruskal–Wallis test indicated that there was no statistically significant difference between all four groups and the positive control at 1, 2 and 3 days (*p* > 0.05), confirming that the cell growth was comparable in the wells with and without the nanotube sheets. In addition, we found significant statistical differences between 1 and 3 days and 2 and 3 days for all the groups and the positive control (*p* < 0.05), with higher cell growth at 3 days, verifying that the coated and non-coated nanotube sheets did not exhibit any cytotoxic effect on the pre-osteoblasts (Figure 3).

As demonstrated in the Figure 4, SEM images show a higher surface coverage by the osteoblasts after 3 days, when observed in 100× magnification. At 2000× and 10,000× magnification, cell morphology can be observed extending throughout the nanotube surface. Cells appeared to be flattened and there was no noticeable difference in the phenotypes observed between different diameters or surface treatments. Those findings corroborate with our cell viability results, demonstrating the biocompatibility and non-cytotoxicity of the non-coated and SiC-coated surfaces on the cells.

### 3.4. Mineralization

MC3T3 cell mineralization was evaluated by the absorbance of the Alizarin-Red staining solution after 21 days of induced osteogenic differentiation. This portion of the study was used to reinforce the evidence of biocompatibility and non-cytotoxicity of both nanotube diameters (50 and 100 nm), as well as the effect of coated and non-coated surfaces. The Kruskal–Wallis test indicated no statistically significant difference between SiC-coated 50 nm and SiC-coated 100 nm, or between non-coated 50nm and non-coated 100 nm for osteoblast calcification (*p* > 0.05). However, significant differences were found between coated and non-coated 50 nm groups, and between coated and non-coated 100 nm groups (*p* < 0.05). These data suggest that the SiC coating contributed to an increase in osteoblasts’ mineralization on tested titanium nanotube surfaces (Figure 5).

## 4. Discussion

There has been great interest in improving the surface of titanium implants to obtain long-term success in clinical applications. In this study, we evaluated the potential of a SiC-coated nanostructured surface and evaluated which nanotube diameter was more conducive for cell proliferation and mineralization of pre-osteoblasts. We found similar results for both 50 nm and 100 nm nanotube diameters, regardless of the surface treatment for both cell proliferation and mineralization. Coated and non-coated nanotube surfaces demonstrated similar biocompatibility and non-cytotoxicity. However, SiC-coated surfaces’ results suggest that these coatings can induce osteoblast mineralization. 

The response of the surrounding tissue depends on the biocompatibility of the material [27]. Surface properties can play an important role in the biological acceptance of the implants [28]. An essential condition for osteoconduction is the direct, stable and extensive contact between bone and the implanted material [27]. One way to optimize the biocompatibility of an implant could be to coat the surface with a biocompatible coating [18]. The biocompatibility can also be improved when tissue can grow into pores of the implanted material [27].

Several studies reported that the surface modification of titanium by anodization (nanostructured surface) can improve osseointegration [4,29]. Zhao et al. [9] found that pre-osteoblasts demonstrated a flatter morphology, extended more filopodia and proliferated faster when cultured on nanostructured surfaces compared with non-nanostructured surfaces. Brammer et al. [11] also had better results for cell proliferation and mineralization on nanotubed or anodized surfaces when compared to non-anodized surfaces. The biocompatibility and non-cytotoxicity of nanostructured titanium surfaces were also demonstrated by several studies [5,9,18].

In our study, no significant changes in cell proliferation were found for different nanotube diameters. Similar results were reported by Voltrova et al. [10], studying osteoblast response on titanium nanotubes with 24, 43 and 66 nm. Brammer et al. [11] compared the cell adhesion, morphology and osteogenic functionality of osteoblasts cultured on titanium nanotubes with 30, 50, 70 and 100 nm and found no difference between diameters for cell proliferation. However, they noticed higher cellular elongation after 24 h for 70 and 100 nm samples by SEM evaluation. These findings can be explained by the increased biocompatibility of the nanostructured surfaces.

Silicon has been found to be an essential element for the growth of bone and cartilage [30]. Our findings demonstrated that the SiC coating was biocompatible and non-cytotoxic; however, the coating did not play a role in cell proliferation. Camargo et al. [8] demonstrated that SiC-coated and uncoated titanium surfaces had improved adhesion of human osteoblasts. They showed that the cell coverage area after 24 h in culture was similar in both the coated and uncoated samples. Zhao et al. [9] also found that a Si coating on titanium nanotube surfaces did not present any negative effect in cell viability of pre-osteoblasts. Wang et al. [30] demonstrated that the Si coating could stimulate the cells to proliferate faster in a 7-day culture when compared to non-anodized titanium and anodized titanium samples. However, authors found that, after 14 days, there was no difference among groups.

Although there was no difference in cell proliferation for either coated or non-coated samples, the mineralization activity of cells was markedly improved when they were cultured on coated compared to non-coated surfaces. Zhao et al. [9] found exactly the same results when they compared titanium nanotube samples coated with Si to non-coated and flat titanium samples. Authors also performed a pull-out test after 2 weeks of implantation in the rat femur and found that the fixation strength was 54% for silicon-coated samples and 18% for non-coated samples. 

Bone mineralization occurs when an inorganic substance, such as calcium, precipitates in an organic matrix such as an osteoblast [4]. Wang et al. [30] affirmed that the silicon coating stimulates osteoblast differentiation and mineralization through upregulation of receptor-related protein 5 and downregulation of Dickkopf-related protein 1 at RNA level.

Furthermore, we found that the SiC coating affected the wettability of nanotube surfaces. Cellular behavior is greatly affected by the wettability of a material [30]. MC3T3 osteoblasts were more likely to adhere to hydrophilic surfaces according to Toffoli et al. [31] According to Ghezzi et al. [32], hydrophilic surfaces enhanced MC3T3 osteoblast adhesion and induced differentiation. Osseointegration occurs in the later stages of the progression of implant fixation, characterized by the establishment of bone–implant contact and peri-implant bone development following bone matrix mineralization [33]. Titanium implant integration into bone is a multistep process involving cell adhesion, growth and differentiation, followed by extracellular matrix production and mineralization [10]. Our results are promising because we demonstrated that SiC coatings on titanium nanotube surfaces substantially enhanced the osteogenic potential of titanium nanotube surfaces, further improving the osseointegration process. XPS analysis has previously revealed chemical composition and SiC uniformity on titanium surfaces [17,26,34]; however, future studies should explore other properties of nanostructured surfaces. Future studies should also explore the mechanisms involved with these outcomes as well as perform in vivo studies to confirm our findings.

## 5. Conclusions

Titanium nanotubes of 50 and 100 nm demonstrated biocompatibility and non-cytotoxicity with pre-osteoblast cells. The nanotube diameter did not affect the pre-osteoblast cell proliferation or differentiation. The SiC coating demonstrated biocompatibility and non-cytotoxicity and it seemed to improve osteoblast differentiation and mineralization on titanium nanostructured surfaces.

## Figures and Tables

**Figure 1 jfb-13-00247-f001:**
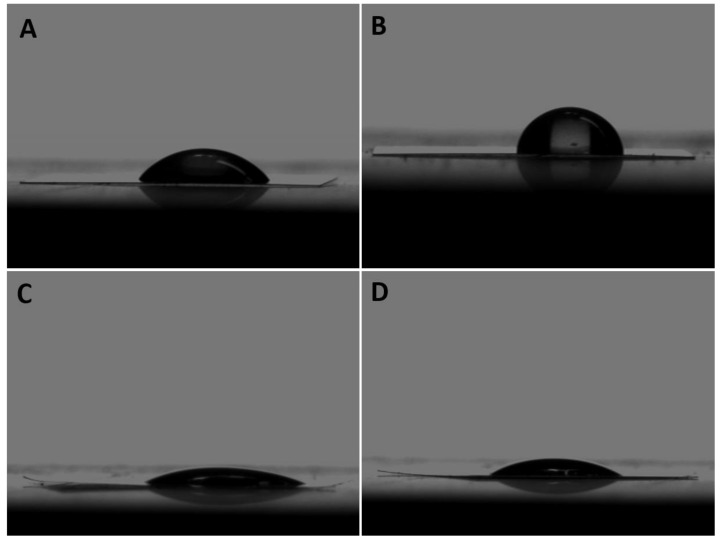
Contact angle images of (**A**) non-coated 50 nm; (**B**) non-coated 100 nm; (**C**) coated 50 nm; and (**D**) coated 100 nm.

**Figure 2 jfb-13-00247-f002:**
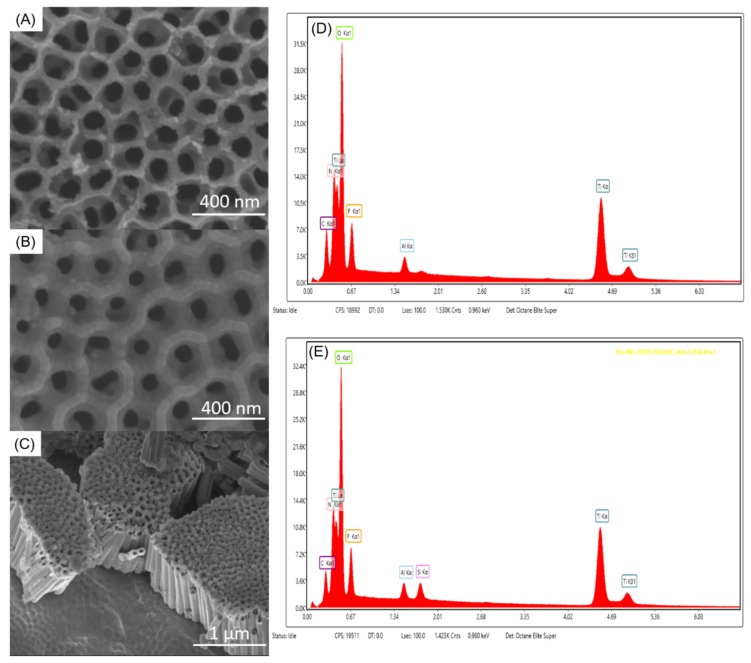
Sample characterization images (reprint from Hsu et al., 2021 [26]). SEM images of (**A**) 100 nm ATO nanotubes, (**B**) SiO_2_/SiC/150 nm ATO nanotubes, (**C**) bending SiO_2_/SiC/ATO nanotubes. Representative EDX spectrums of (**D**) non-coated ATO nanotubes and (**E**) SiO_2_/SiC-coated ATO nanotubes.

**Figure 3 jfb-13-00247-f003:**
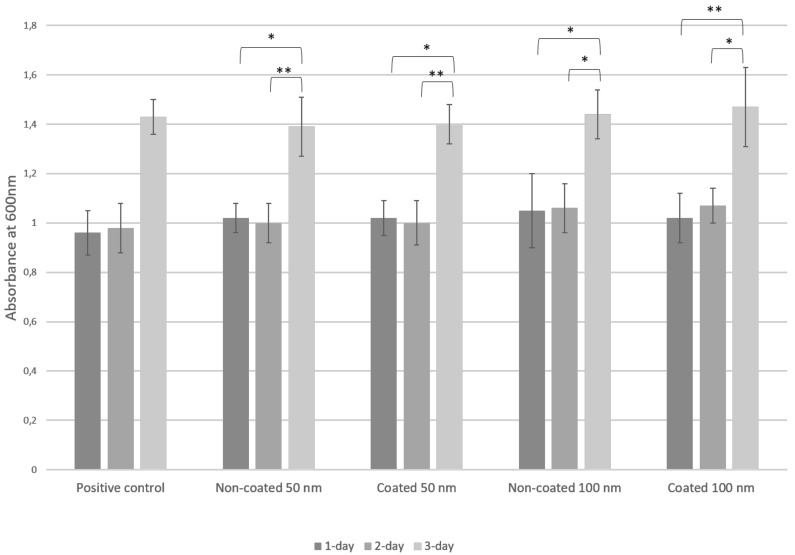
Cell viability of MC3T3 pre-osteoblasts assessed by absorbance of CellTiter-Blue assay for positive control and experimental groups after 1, 2 and 3 days. Asterisks indicate statistical significance analyzed by Kruskal–Wallis test (* < 0.05; ** = 0.000).

**Figure 4 jfb-13-00247-f004:**
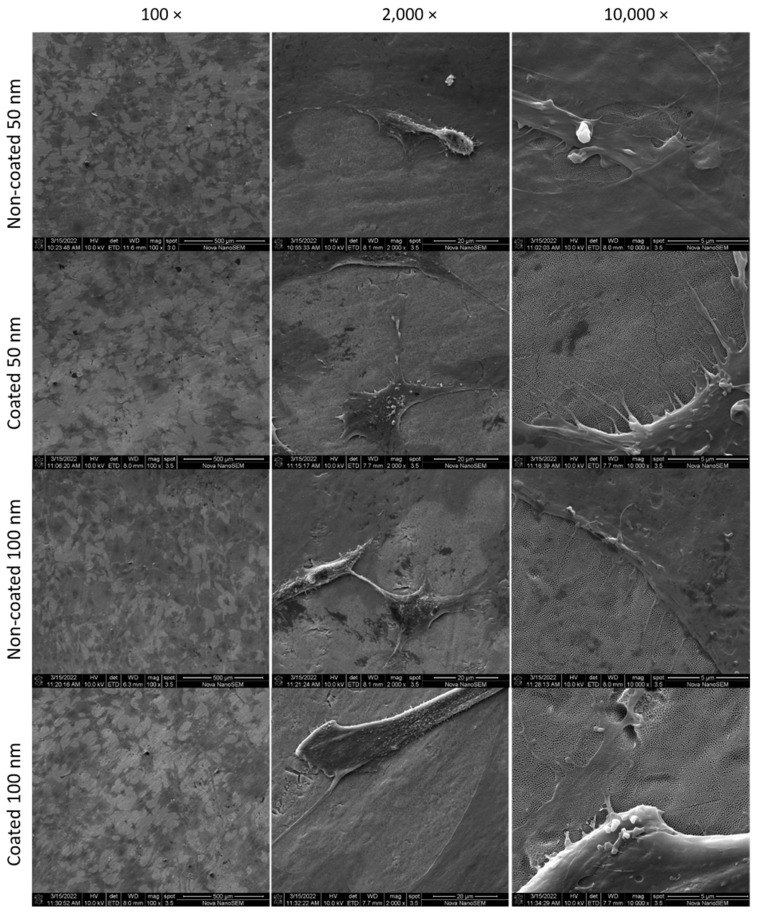
SEM images showing MC3T3 pre-osteoblasts on titanium nanotube sheets after 3-day culture. The darker areas in 100× images were the surfaces covered by the cells.

**Figure 5 jfb-13-00247-f005:**
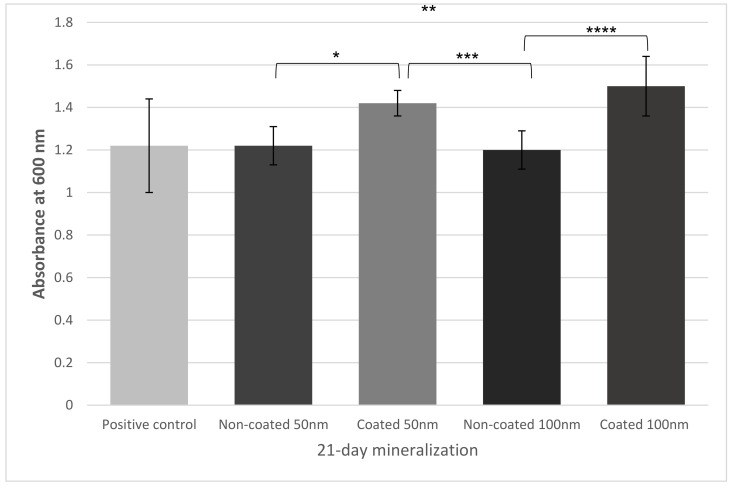
Mineralization of MC3T3 osteoblasts assessed by absorbance of Alizarin-Red staining solution for positive control and experimental groups after 21 days. Asterisks indicate statistical significance analyzed by Kruskal–Wallis test (* = 0.014; ** = 0.002; *** = 0.006; **** = 0.001).

**Table 1 jfb-13-00247-t001:** Contact angle measurement.

Group	Mean Contact Angle ±SD
Non-coated 50 nm	43.7 ± 4.6°
Non-coted 100 nm	64.4 ± 0.3°
Coated 50 nm	23.5 ± 3.8°
Coated 100 nm	29.2 ± 4.8°

## Data Availability

Data available upon request.

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
