# Peer review of "Effect of Silicon Carbide Coating on Osteoblast Mineralization of Anodized Titanium Surfaces"

_jfb, 2022, doi:10.3390/jfb13040247_

Round 1

Reviewer 1 Report

In this manuscript, the authors studied the effect of titanium nanotube diameter and silicon carbide (SiC) coating on the proliferation and mineralization of pre-osteoblasts. They found that titanium nanotube diameters (50 nm and 100 nm) have no impact on the cell viability or mineralization of pre-osteoblasts on either SiC coated or non-coated titanium nanotube sheets. Even though SiC coating did not affect cell viability, it promoted mineralization of pre-osteoblasts on titanium nanostructured surface, compared to non-coated groups. 

The findings are consistent with those from previous similar studies. Here are several minor concerns:

1) It is not clear why the authors only tested the effects of the nanotube diameters of 50 nm and 100nm, but not the effects of any other sizes of nanotube diameters. 

2) Line 118 on page 3: "Cells were seeded at 2x104 cells/ml and "

Line 134 on page 3: "A quantity of 2x104 cells /well were" 

Comment: It is not clear if the same number of cells were plated in each well of 24-well plate for all the experiments.

3) It is highly recommended that the authors should make notes about the statistical results in the Figures and Figure legends. 

Author Response

Manuscript JFB-1865957

Aug 15, 2022

We would like to thank the reviewers for their valuable and constructive criticisms. We have responded to all items as follow. All suggestions and corrections were made into the text using the track changes of Microsoft office word. We also reformatted the references and citations.

Sincerely yours,

Josephine Esquivel-Upshaw (corresponding author)

COMMENTS AND ANSWERS – REVIEWER 1

  • It is not clear why the authors only tested the effects of the nanotube diameters of 50 nm and 100nm, but not the effects of any other sizes of nanotube diameters.

We explained this on lines 61, 62, 96 and 97 (page 2) of the manuscript. We wanted to test two extremes for diameters and determine which one was most conducive to bone growth.

2) Line 118 on page 3: "Cells were seeded at 2x104 cells/ml and "

Line 134 on page 3: "A quantity of 2x104 cells /well were"

Comment: It is not clear if the same number of cells were plated in each well of 24-well plate for all the experiments.

We used the same number of cells in each well, we reformulated to make it clear on lines 129 and 149 (pages 3 and 4).

3)It is highly recommended that the authors should make notes about the statistical results in the Figures and Figure legends.

We agree with the reviewer and added as suggested for Figures 1 and 3 (pages 5 and 7).

COMMENTS AND ANSWERS – REVIEWER 2

  1. Materials and methods: 2.1.; No information about the anodizing conditions of titanium substrates used in all experiments?

The nanotube sheets were pre-fabricated and we didn’t control the anodization process, we add this information on lines 93 to 95 (page 2) to make it clear.

  1. Results; Why were TNT layers with only 50 and 100 nm diameters investigated? It would be important to investigate TNT layers with smaller diameters.

We explained this on lines 96 and 97 (page 2). As mentioned earlier, we wanted to test extremes in diameter sizes. Additionally, anodization usually produces a range of diameters with +/- 20um.

What is the hydrophobicity of the TNT coatings tested?

Contact angle measurements were conducted on the different coatings. We added this data on lines 116 to 122 (page 3), 190 to 197 (pages 4 and 5) and 294 to 298 (page 8).

No data concerning the morphology and the structure of studied TNT coatings and TNT + SiC systems. The XRD, EDX, XPS data should be added and discussed.

We performed these analyses extensively and reported them in previous studies. (Fares C, Hsu SM, Xian M, Xia X, Ren F, Mecholsky JJ, et al. Demonstration of a SiC Protective Coating for Titanium Implants. Materials. 2020 Jul 26;13(15):3321; Fares C, Elhassani R, Partain J, Hsu SM, Cracium V, Ren, F, Esquivel-Upshaw J. Anneling and N2 plasma treatment to minimize corrosion of SiC-coated glass-ceramics. Materials. 2020; 13: 2375).

  1. Conclusions; In this part of the manuscript, the authors summarized the results obtained.No final conclusions.

We reformulated the conclusion to answer our aims that were (1) to analyze the influence of titanium nanotube diameter and (2) the potential of SiC coating on cell proliferation, and mineralization. (lines 313 to 315 on page 9)

Reviewer 2 Report

The authors of the reviewed manuscript present the research results concerning evaluating the influence of the titanium nanotube diameter and the effect of silicon carbide (SiC) coating on the proliferation and mineralization of pre-osteoblasts on titanium nanostructured surfaces. The anodized titanium sheets with nanotube diameters of 50 and 100nm were used in these investigations. In the conclusions, the authors note that there is no influence of the nanotube diameter on cell viability or pre-osteoblast mineralization on SiC-coated or uncoated titanium nanotube sheets. Simultaneously, they noticed the increase of the osteoblast mineralization on the titanium nanostructured surfaces enriched by SiC layer.

The paper requires the following amendments:

1. Materials and methods: 2.1.; No information about the anodizing conditions of titanium substrates used in all experiments?

2. Results; Why were TNT layers with only 50 and 100 nm diameters investigated? It would be important to investigate TNT layers with smaller diameters.

What is the hydrophobicity of the TNT coatings tested?

No data concerning the morphology and the structure of studied TNT coatings and TNT + SiC systems. The XRD, EDX, XPS data should be added and discussed.

3. Conclusions; In this part of the manuscript, the authors summarized the results obtained. No final conclusions.

In my opinion, after major revision, this manuscript may be accepted for publication in the Journal of Functional Biomaterials.

Author Response

(The authors gave the same response as above.)
